# Effects of Injectable Administration of Dexamethasone Alone or in Combination with Vitamin E/Se in Newborn Low Birth Weight Piglets

**DOI:** 10.3390/vetsci10020135

**Published:** 2023-02-09

**Authors:** Georgios I. Papakonstantinou, Dimitrios A. Gougoulis, Nikolaos Voulgarakis, Georgios Maragkakis, Dimitrios Galamatis, Labrini V. Athanasiou, Vasileios G. Papatsiros

**Affiliations:** 1Clinic of Medicine, Faculty of Veterinary Medicine, School of Health Sciences, University of Thessaly, 43100 Karditsa, Greece; 2Department of Animal Science, University of Thessaly, 41110 Larissa, Greece

**Keywords:** low-birth-weight piglet, piglet health, dexamethasone, vitamin E/Se

## Abstract

**Simple Summary:**

This study aimed to evaluate the effects of intramuscular administration (IM) of dexamethasone (Dexa) alone or in combination with Vit E/Se on LBW piglets during the early postnatal period. 100 LBW piglets were divided into 5 groups and treated with IM Dexa alone or in combination with Vit E/Se after birth: (a) Group A: Control group, (b) Group B: Dexa on D1 (1st day of life), (c) Group C: Dexa on D1, D2, D3, (d) Group D: Dexa + Vit E/Se on D1, and (e) Group E: Dexa + Vit E/Se (IM) on D1, D2, D3. A significant increase in piglets’ BW and ADWG in Group E and a reduction in Group C were noticed, respectively. Vitality scores were lower in piglets of Group B-Dexa1 and Group C-Dexa3, respectively. Furthermore, piglets of Group C showed poor clinical performance. In conclusion, the administration of a Dexa and Vit E/Se combination shortly after the birth of LBW piglets for 1–3 days could enhance their growth and their ongoing productivity.

**Abstract:**

Increasing litter size may lead to low-birth-weight piglets (LBW) and further negative long-term effects. This study aimed to evaluate the effects of intramuscular administration (IM) of dexamethasone (Dexa) alone or in combination with vitamin E/Se on LBW piglets during the early postnatal period. The study included a total of 100 LBW piglets that were divided into 5 groups (20 LBW piglets per group) and treated with IM Dexa alone or in combination with vitamin E/Se (Vit E/Se) after birth as follows: (a) Group A-Cont: Control group, (b) Group B-Dexa1: Dexa on D1 (1st day of life), (c) Group C-Dexa3: Dexa on D1, D2, D3 (D2: 2nd day of life, D3: 3rd day of life), (d) Group D-Dexa + VitE/S1: Dexa + Vit E/Se on D1, and (e) Group E-Dexa + VitE/S3: Dexa + Vit E/Se (IM) on D1, D2, D3. Body weight (BW) and the Average Daily Weight Gain (ADWG) were recorded for all piglets on days 1, 7, 14, and 25, and vitality score (VS) was recorded on days 1, 2, 3, 4, and 14. A significant increase in BW and ADWG in Group E-Dexa + VitE/S3 and a significant reduction in Group C-Dexa3 were noticed in comparison to other groups. VS in groups Group B-Dexa1 and Group C-Dexa3 were significantly lower in comparison to other groups. Furthermore, piglets of Group C-Dexa3 had a significantly higher frequency of clinical findings compared to other groups. In conclusion, the administration of Dexa and vitamin E/Se combined after the birth of LBW piglets for 1–3 days has beneficial effects on their growth and survival scores.

## 1. Introduction

Despite the technological innovations and management improvements in pig farming, the mortality of suckling piglets remains a major economical and welfare concern. The annual cost of neonatal mortality in the swine industry is high [1]. Most neonatal mortality occurs in the first days, especially in the first 72 h of life [2], reflecting the problems of transition from a protected intrauterine life to extrauterine existence [3]. However, the mortality of suckling piglets is a multifactorial problem, including maternal and piglet-related factors, excluding infectious causes [3]. The main causes of death during lactation are low-birth-weight (LBW) piglets, starvation, crushing of sick piglets, and diarrhea [4,5]. Perinatal mortality is particularly high in LBW piglets [6,7,8,9]. A decrease in the piglet birth weight increases the risk of a higher preweaning mortality rate [10], as only 28% of piglets weighing less than 1.1 kg at birth survived to 7 days [3]. The use of hyperprolific sow lines has increased litter size considerably in the last decades. The increased litter size results in a larger number of piglets than the available sow’s teats as well as in an increased birth-weight variation within the litter, characterised by an increased number of LBW piglets with reduced vitality; all of these increase the piglets’ competition for colostrum intake and have a negative impact on piglets’ survival [11,12,13,14]. Inadequate or no colostrum intake results in piglet starvation, rendering them prone to diarrhea and crushing [15]. Hypothermia is a major cause of mortality in neonatal piglets. Perinatal mortality in piglets, especially the first days after birth, is frequently caused by non-infectious conditions, such as hypoglycemia or low birth weight, which can be associated with hypothermia experienced at birth [9]. Hypothermia can be a significant cause of death in newborn piglets, and although this condition is not infectious, it is considered an important factor in preweaning mortality [16,17]. The piglet’s ability to overcome postnatal hypothermia during the immediate postpartum period is directly related to birth weight and its position among sow and littermates during the first 2 h after birth [18,19]. Birth weight is the most important single factor in successful recovery from postnatal hypothermia [19]. In addition, having lower energy stores make them more sensitive to cold than normal-weight piglets. Furthermore, LBW piglets are at long-term risk from infectious diseases, as they fail to acquire sufficient immunity from colostrum because of delayed or limited suckling [15,20]. 

Colostrum contains mainly IgG, as well as IgA and IgM, leukocytes, selenium, and vitamin E, all of which are important for immune function [21]. Vitamin E is a lipid-soluble vitamin, with α-tocopherol being the most biologically active form and calculated as about 90–100% of the vitamin E found in tissue [22]. Vitamin E is the principal chain-breaking antioxidant in body tissues and plays the main role in the defense against lipid peroxidation, protecting cell membranes at an early stage of free radical attack [23]. Deficiency of vitamin E in the diet can induce damage to cell membranes, including immune cells [23]. In the newborn piglet, vitamin E status increases after the suckling of colostrum, but plasma and tissue levels of α-tocopherol remain low, which is the result of low-rate transfer across the placenta [23]. Due to inadequate colostrum intake, LBW piglets are more exposed to the risk of oxidative stress and immune shielding provided by vitamin E deficiency [23]. Selenium (Se) is a valuable trace element for the regulation of immunity functions [24] and the growth performance of newborn piglets [25,26]. In LBW piglets, reduced colostrum and, later, milk intake leads to Se deficiency, resulting in the induction of oxidative stress and susceptibility to various pathogens [27,28].

Glucocorticoids play an important role in intestinal maturation and function [29], in the regulation of prenatal and postnatal growth [30], metabolism [31], and homeostasis [32]. Furthermore, studies suggest that the surge in glucocorticoids, which are related to the natural birth process, is a significant mediator of postnatal development and growth in viviparous animals [33].

This study aimed to evaluate the effects of intramuscular (IM) administration of a synthetic glucocorticoid (dexamethasone) in combination with vitamin E/Se on LBW piglets during the early postnatal period in an endemically PRRSV-infected farm. Due to the noncommercial availability of products exclusively with vitamin E or Se in the Greek market, a study was conducted on their combined effect.

## 2. Materials and Methods

### 2.1. Trial Farm

The study was carried out on a farrow-to-finish commercial pig farm with a capacity of 400 sows’ production (commercial hybrids of Large White × Landrace), located in southern Greece, from April to May 2021. Based on the results of routine blood sampling (breeding stock, weaning, growing, and finishing stage) the farm was PRRSV-infected. The results revealed positive samples in growers and finishers (90–160 days of age) for PRRSV type 1 (European genotype) using Real-Time Polymerase Chain Reaction (RT PCR). The clinical picture of animals was characterized mainly by respiratory symptoms including coughing, pyrexia, poor performance, and increased mortality due to secondary infections.

All sows of the farm were vaccinated against Aujeszky’s disease virus, parvovirus, atrophic rhinitis, erysipelas (Erysipelothrix rhusiopathiae), PRRSV (Modified Lived Vaccine-MLV), Escherichia coli, and Clostridium perfringens infections. Weaners were vaccinated against Porcine Circovirus type 2 and Mycoplasma hyopneumoniae. The PRRSV MLV vaccination of sows was performed as a mass vaccination every 3 months. Control of endo/ectoparasites was currently maintained by sows’ treatment with a single ivermectin injection 14 days before farrowing.

### 2.2. Experimental Material

A commercial formulation of injectable dexamethasone of 2 mg/mL concentration (as 2.63 mg dexamethasone sodium phosphate) was administered IM in LBW piglets at a dose of 0.03 mL/Kg BW or 0.06 mg/Kg BW (Dexamethasone; Provet). The experimental groups are presented in Table 1. A commercial product of an injectable combination of vitamin E and Se (vitamin Ε 150 mg/mL and Se 1.67 mg/mL) was applied IM in LBW piglets at a dose of 0.03 mL/Kg BW (Vitamin E-Selen; MSD Animal Health). The piglets in group A (control group) received IM Sodium Chloride 0.9% (Vioser S.A.).

### 2.3. Study Procedure/Animals

The trial started 7 days after the last mass vaccination of breeding stock against PRRSV with an MLV vaccine. In total, one hundred (100) newborn LBW piglets were selected from 100 sows (50 sows with parity 1 and 50 sows with parity 2) during the first 0–9 h after the birth of the last piglet in each farrowing. Each selected sow had at least 14 (liveborn) piglets based on the farm records. Randomization was applied by the sealed envelope system when a sow with at least 14 piglets was presented. For each selected sow, a total of 14 piglets were kept with the sow, while no cross-fostering was applied. LBW piglets were selected (1 piglet per litter) from newborn piglets on the first day of life based on their BW of less than 1 kg (range 0.6–1.0 kg). The selected LBW piglets were equally distributed based on their sex and the parity of their mother; the selected male piglets were not castrated. The weaning age was at 25 days. The large farm capacity allowed us to equally distribute the underweight piglets based on their sex (56 males and 44 females).

Each farrowing room included pens with farrowing crates, including nipple drinkers and separate removable feeders for the sows and the piglets. The drinking water was provided automatically, and the flow of the nipples was checked every day by an animal technician. The farrowing rooms were equipped with a fully automated feeding system and a climate monitoring system for temperature and humidity. The farrowing rooms were maintained at ambient temperature (23 ± 0.5 °C) with lights on/off at 07:00/21:00, and natural light was provided by windows in each room. An infrared heat lamp was suspended in the center of the floor area on one side of the farrowing crate over an insulated rubber mat (the average temperature under the heat lamp during the study period was approximately 30–35 °C). All piglets were kept in farrowing pens under the same conditions. The large farm capacity allowed us to complete the study over a period of 3 months with approximately the same conditions throughout the study.

The selected one hundred (100) newborn LBW piglets were divided into 5 groups (20 LBW piglets per group) and treated intramuscularly (IM) with dexamethasone (Dexa) alone or in combination with vitamin E + Se (Vit E/Se) according to Table 1. Piglets in the control group received a placebo (Sodium Chloride 0.9%; Vioser S.A.) dose of 0.03 mL/Kg BW. Each pig was injected in the right neck muscle intramuscularly, using an automatic syringe with a 20 gauge x ½ inch needle.

### 2.4. Records

The following parameters were recorded for each piglet: (a) body weight (BW) on D1, D7, D14, and D25 (weaning day) after birth, (b) vitality score (VS) on D1, D2, D3, D4, and D14 after birth, and (c) clinical observations, according to Table 2. VS was based on a modified scoring system, using Randall’s [34] adaptation of the Apgar score for human neonates described by Zaleski and Hacker [35] and modified by Okere et al. [36] and Mota-Rojas et al. [37], measuring the variables according to Table 2. A stethoscope was used to monitor the heart rate of all piglets. The data were recorded by the same person. The vitality score and clinical observations of each piglet were scored continuously by the observer for the first 15 min of every hour of data collection. The observer was blinded as to time point, litter, and piglet treatment. 

The day of the first intervention (D1) for LBW piglets was the day of the farrowing (estimating time 0–9 hours after the birth of their last littermate). On the days of administration of Dexa alone or in combination with Vit E/Se, LBW piglets were placed in a transport cart and administered their assigned treatments. Separation of the LBW piglets from their sow and littermates lasted ~2–3 min. LBW piglets of each group were weighed using a digital weighing machine during each time point of the trial period. Separation of the LBW piglets from their sow and littermates lasted ~5–10 min.

### 2.5. Sampling and Laboratory Examinations

Blood samples were collected for laboratory examinations [Real-time polymerase chain reaction (RT PCR) and ELISA] from 5 piglets from each group on the 7th, 14th and 25th day (weaning day) of their age. A total of 15 blood samples were collected from each trial group (5 piglets ∗ 3 blood samples obtained from each one = 15).

All blood samples were examined by RT PCR for PRRSV-1 and PRRSV-2 infection. RNA extracts were examined by RT PCR for PRRSV-1 (EU type 1 and NA type 2). Viral RNA was purified from all serum samples by using the RNeasy Mini kit (Qiagen, Germany) in an automatic robot (QIAcube; Qiagen) following the manufacturer’s instructions. The entire procedure was performed in Laboratories Hipra S.A. (Amer, Girona, Spain).

### 2.6. Statistical Analysis

All data were collected and processed in Excel 2013 (Microsoft Office 2013). For body weight (BW) analysis, the General Linear Model for Repeated measures Analysis of Variance (ANOVA) was selected. The normality test on day 1 of the trial was applied using a Shapiro-Wilk Test. All comparisons of each possible pair of groups were conducted using post-hoc testing with Bonferroni correction. Time points of collecting data were selected as Within-Subjects Variables and Group allocation as Between-Subject Factors.

Vitality score (VS) analysis was based on a five-tiered scale of ordinal data (Table 2). Due to the number of groups (>2), the Kruskal-Wallis H test was selected for the analysis. According to the distribution of each group’s data, the variability was not similar, and therefore the mean rank test was applied for the analysis. Post-hoc analysis was applied for further analysis of the difference. On day 2 of the trial, clinical observations were evaluated using 3 ordinal categories. For the comparison of the clinical findings between groups, chi-square analysis was applied in a 5 × 3 table and multiple comparisons were conducted for each possible pair of groups. Statistically, the significant level was set to 0.05. The SPSS statistical package was used for data analysis and presentations (IBM Version 20).

## 3. Results

### 3.1. Mortality

All data were normally distributed according to a Shapiro-Wilk Test on day 1. Three animals died on day 7 and two animals on day 14 of the trial. One of the five animals died due to crushing from the sow, and the remaining four died due to severe diarrhea based on pathological findings of necropsy.

### 3.2. Body Weight (BW)

Repeated measure analysis was applied for mean BW analysis. A repeated measures ANOVA with a Greenhouse-Geisser correction determined that mean BW differed statistically significantly between time points (F = 6169.535, *p* < 0.05). 

The Average Daily Weight Gain (ADWG) of each period between recordings is presented in Table 3. A post-hoc test using Bonferroni correction revealed that administration of dexamethasone alone or in combination with vitamin E + Se could affect the ADWG rate over time (F = 7.62, *p* < 0.05). Animals in group C had the lowest ADWG throughout the trial. During the first 7 days of the survey, piglets of Group E had a higher ADWG rate (0.25 kg/day) compared to all other groups, with piglets of group E following with an ADWG of 0.19 kg/day). In the period between days 8 and 14, the ADWG of group E (0.27) was significantly higher compared to groups B and C (0.17 and 0.20 respectively). Details of the ADWG fluctuation during the survey are presented in Table 3.

Similarly, the mean BW of each group on each day of sampling is presented in Table 3. Post-hoc tests revealed that administration of dexamethasone with or without vitamin E + Se combination could affect the BW scoring over time (F = 21.217, *p* < 0.05). The mean difference of BW in animals of groups A compared to animals of groups B-Dexa1 and D-Dexa + VitE/S1 was not statistically significant during the trial (*p* = 0.218 and *p* = 1, respectively). Throughout the trial, animals in group C had a lower BW compared to all other groups, with the mean difference being −0.67, −0.41, −0.76, and −1.1, for A, B, D, and E, respectively. The mean BW of animals in group E was higher compared to all other groups. The mean difference of BW between groups E-Dexa + VitE/S3 and A-Cont was 0.48 with *p* = 0.00, between groups, E-Dexa + VitE/S3 and B-Dexa1 was 0.74 with *p* = 0.00, between Group E-Dexa + VitE/S3 and Group C-Dexa3 was 1.15 with *p* = 0.00, and between groups E-Dexa + VitE/S3 and D-Dexa + VitE/S1 was 0.387 with *p* = 0.03. Figure 1 shows the estimated marginal mean of the BW score of each group on days 1, 7, 14, and 25 (time 1, 2, 3, and 4, respectively). The highest mean BW score from days 1 to 25 was noticed in group E.

### 3.3. Vitality Score (VS)

According to the results, the Kruskal-Wallis H test showed that there was a significant difference in VS between the different treatments at every point of sampling during the trial (for D1 *p* = 0.00, for D2 *p* = 0.00, for D3 *p* = 0.00, for D4 *p* = 0.002, and for D14 *p* = 0.00). In Table 4, the mean rank score for each group on each sampling day is presented. Post-hoc analysis for each sampling point for D1 revealed a significant difference between groups B-Dexa1 and A-Cont (*p* = 0.001) and between groups E-Dexa + VitE/S3 and A-Cont (*p* = 0.001). On D2, groups D-Dexa + VitE/S1 and E-Dexa + VitE/S3 had significantly higher mean rank scores compared with groups B and C (group D vs B *p* = 0.00, group D vs C *p* = 0.013, group E vs B *p* = 0.00 and group E vs C *p* = 0.013). On D3, groups D-Dexa + VitE/S1 and E-Dexa + VitE/S3 had significantly higher VS compared to groups B-Dexa1 and C-Dexa3, and this significant difference was sustained throughout the trial. Analytical data are presented in Table 5.

### 3.4. Clinical Findings 

A chi-square test of independence was performed to examine the relationship between the administration of dexamethasone with/without vitamin E and Se and pigs’ excrements. The relation between these variables was significant [(8, N = 100) = 45, 140, *p* < 0.05]. Piglets of group C-Dexa3 had a significantly different defecation frequency of diarrhea compared to the other groups (group B vs A *p* = 0.001, group B vs C *p* = 0.045, and B vs E *p* < 0.05), as is shown in Figure 2. The higher diarrhea score of 3 was noticed in groups B-Dexa1 and C-Dexa3, while it was not noticed in group E-Dexa + VitE/S3.

### 3.5. RT PCR Results 

All blood serum samples that were examined by RT PCR for PRRSV-1 and PRRSV-2 were negative.

## 4. Discussion

Increasing litter size through genetic selection and management strategies is one of the main issues in the modern pig industry [38,39]. The increased litter size is a major challenge for the modern hyperprolific sows’ physiology during pregnancy, parturition, and lactation, as well as the piglets’ survival [13]. In large litters, the proportion of LBW piglets is increased, resulting in long-term welfare and performance consequences, such as reduced BW and disease viability, as well as increased mortality risk [4,7,20,40,41]. Our study aimed to investigate whether the administration of dexamethasone alone or in combination with vitamin E + Se in newborn LBW piglets could improve their survival in heterogenous litters and enhance their growth in an endemically PRRSV-infected farm. Based on the results of our trial, the single administration of dexamethasone to neonatal LBW piglets (groups B-Dexa1 and C-Dexa3) showed poor effects on their growth. In particular, the piglets that received IM dexamethasone for 3 days (group C-Dexa3) showed a statistically significant reduction in BW and ADWG compared to all other groups. These results contrast with a previous study by Carroll (2021), which has shown that the body weights of piglets that received a single dose of dexamethasone within one hour from birth were not significantly different until day 18 at which time BW was 10.1% greater for the dexamethasone-treated piglets in comparison to the piglets of the control group [33]. Moreover, our results showed that the administration of dexamethasone alone for 1 or 3 days after birth in LBW piglets has no statistically significant effect on BW at weaning day (day 25), in agreement with previous studies [42]. Previous studies demonstrated that the injectable administration of newborn piglets with vitamin E + Se did not affect their growth rates until weaning age [43]. Our trial is the first study that investigated the effects of the injectable administration of dexamethasone and vitamin E + Se in LBW newborn piglets. However, further studies on concentrations, timing, and different combinations (especially with only vitamin E + Se combination administration) are required to maximize the beneficial effects of such interventions.

Furthermore, the groups that received IM dexamethasone for 1 or 3 days (group B-Dexa1 and C-Dexa3, respectively) showed lower vitality in comparison to piglets of the control group and groups that received IM dexamethasone and vitamin E + Se for 1 or 3 days (group D-Dexa + VitE/S1 and E-Dexa + VitE/S3, respectively). Consequently, the most beneficial effect on the growth, vitality, and clinical performance of neonatal LBW piglets was present in group E-Dexa + VitE/S3, which received IM dexamethasone and vitamin E + Se combination for the first 3 days of life. However, in the current study, the colostrum intake was not estimated to evaluate if the LBW piglets received sufficient levels of E and Se from the colostrum, as sufficient colostrum intake is fundamental for their survival [3,9,20,39,44].

Due to inadequate colostrum intake, LBW piglets are more exposed to the risk of oxidative stress [23], due to low levels or deficiency of vitamin E or/and Se [23,27]. Notably for vitamin E, it is well established that vitamin E content increases after the suckling of colostrum, but plasma and tissue levels of vitamin E remain low due to low transfer rates across the placenta [23]. For this reason, the routine IM administration of dexamethasone in combination with vitamin E/Se in LBW piglets for the first 3 days of life could be proposed, aiming to increase tissue vitamin E + Se concentration by preventing oxidative stress and resulting in an improved performance [45]. As neonatal piglets suffer from serious oxidative stress because of their immature antioxidant system, the increase in vitamin E levels in LBW piglets by IM administration could help them to recover from the negative effects of oxidative stress [46,47]. Vitamin E and Se are both the major chain-breaking antioxidants in body tissues and are considered the first line of defense against lipid peroxidation, protecting cell membranes at the early stages of free radical attack thanks to their free radical scavenging activity [46,47,48,49]. Recently, Wang et al. [50] reported that the supplementing the maternal diet of sows with vitamin E at a high concentration improved the body weight of piglets at weaning and enhanced humoral immune function and antioxidant activity in sows and piglets. 

As for the IM administration of both dexamethasone and vitamin E/Se combination in our study at the first 3 days of life, the use of dexamethasone could improve the growth of LBW piglets due to its beneficial effects on their digestive capability, metabolism, and reaction to stress [29,33,51]. In a recent study, it was reported that the use of dexamethasone in piglets infected by highly pathogenic PRRSV (HP-PRRSV) increases the disease severity and should be avoided in the clinical treatment of HP-PRRS [52]. In our study at an endemically PRRSV-infected farm, similar results were noticed using dexamethasone, as a significant reduction in growth performance and a high frequency of clinical findings were noticed in LBW piglets with IM administration of dexamethasone alone for the first 3 days of life. However, we did not notice significant differences in PRRSV Abs in suckling piglets between groups at the 7th, 14th, and 25th day of age. Furthermore, significantly lower VS was noticed in LBW piglets that received IM dexamethasone alone for the first day or the first 3 days of life.

LBW piglets that received only dexamethasone showed poorer growth compared to piglets that received an IM combination of dexamethasone and vitamin E + Se. However, the potential effects of LBW piglets receiving only a vitamin E + Se combination during our study were not investigated. So, in future research on the prospects for improving the growth and health rates of LBW neonatal piglets, investigating the administration of just vitamin E + Se combination would be of great interest.

## 5. Conclusions

Increasing litter size is one of the main challenges of the global pig industry. However, many studies have reported that the large litter size is associated with the birth of many LBW piglets and a high mortality rate. Proper management could eliminate the negative implications for animal welfare, resulting in better growth and health scores. According to our study, there are encouraging results in reducing preweaning mortality rate in sows with increased litter size. The IM administration of dexamethasone with vitamin E + Se combination shortly after the birth of piglets and for 1–3 days could enhance their growth rate. Therefore, this practice as a routine program in pig farms with increased litter size could be a useful tool to reduce preweaning mortality and consequently reduce the cost of production. However, further research on the dosage and timing of administration of dexamethasone with vitamin E + Se combination should be conducted to maximize the beneficial effects of such interventions.

## Figures and Tables

**Figure 1 vetsci-10-00135-f001:**
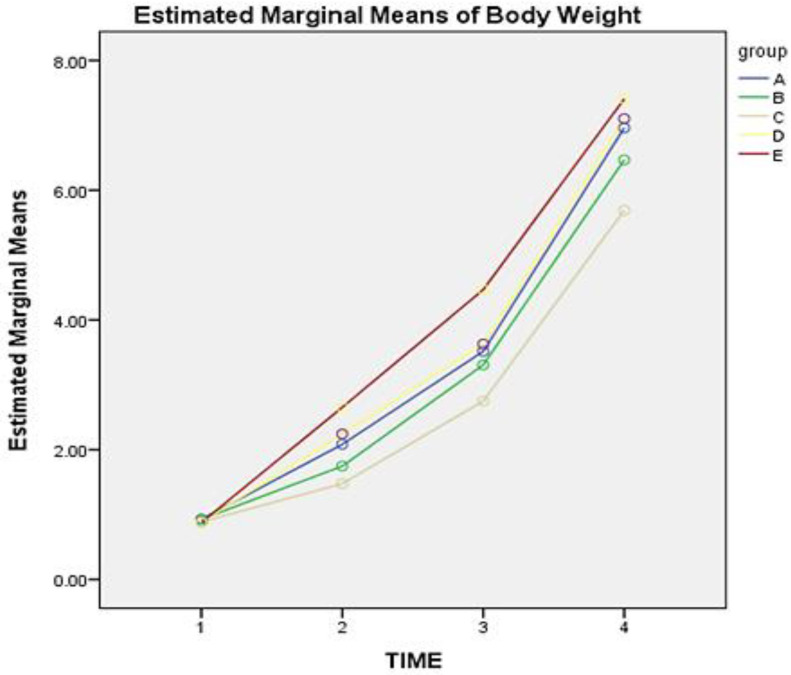
Mean of body weight score of piglets in groups A, B, C, D, and E on days 1, 7, 14, and 25 (time 1, 2, 3, and 4 respectively). Group A-Cont (control group). Group B-Dexa1 (piglets treated with IM dexamethasone on D_1_). Group C-Dexa3 (piglets treated with IM dexamethasone on D_1_, D_2_, and D_3_). Group D-Dexa + VitE/S1 (piglets treated with IM dexamethasone and vitamin E/Se on D_1_). Group E-Dexa + VitE/S3 (piglets treated with IM dexamethasone and vitamin E/Se on D_1_, D_2_, and D_3_).

**Figure 2 vetsci-10-00135-f002:**
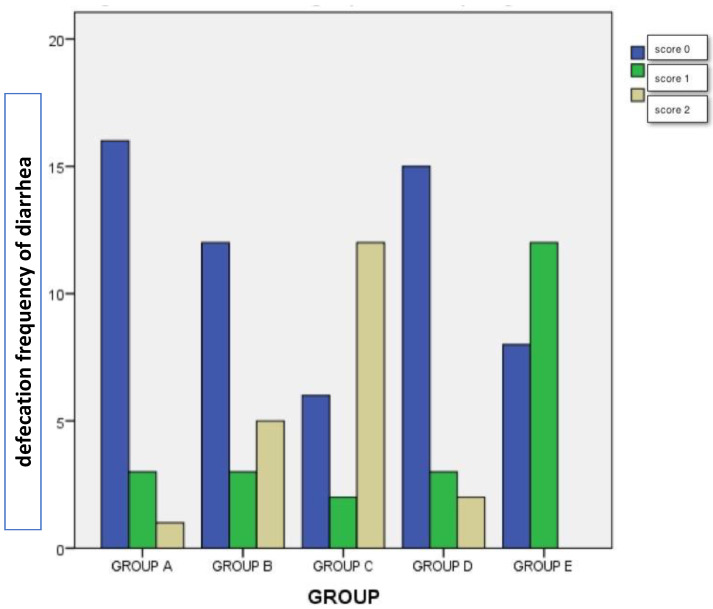
The defecation frequency of diarrhea (scoring 0 to 3) in groups A, B, C, D, and E. Group A-Cont (control group). Group B-Dexa1 (piglets treated with IM dexamethasone on D_1_). Group C-Dexa3 (piglets treated with IM dexamethasone on, D_1_, D_2_, and D_3_). Group D-Dexa + VitE/S1 (piglets treated with IM dexamethasone and vitamin E/Se on D_1_). Group E-Dexa + VitE/S3 (piglets treated with IM dexamethasone and vitamin E/Se on D_1_, D_2_, and D_3_).

**Table 1 vetsci-10-00135-t001:** Experimental groups of the trial.

Group	Treatment
Group A-Cont	Control group-Cont
Group B-Dexa1	IM Dexamethazone on D_1_
Group C-Dexa3	IM Dexamethazone on D_1_, D_2_, D_3_
Group D-Dexa + VitE/S1	IM Dexamethazone on D_1_ + IM Vitamin E/Se on D_1_
Group E-Dexa + VitE/S3	IM Dexamethazone D_1_, D_2_, D_3_ + Vitamin E/Se D_1_, D_2_, D_3_

D_1_: Day 1-1st day of life, D_2_: Day 2-2nd day of life, D_3_: Day 3-3rd day of life.

**Table 2 vetsci-10-00135-t002:** Criteria for clinical observations and vitality score (VS).

Clinical Observations					
Score of General Behavior		Score of Suckling		Score Of Gastrointestinal Signs	
0-No abnormalities		0-Appear normal		0-Appear normal	
1-Mild depression, reluctance to move		1-Hardly interested in suckling		1-Pasty feces or watery mild yellow diarrhea	
2-Reduced general condition, extended resting		2-Without clear suckling		2-Watery moderate yellow diarrhea or reddened anal region	
3-Strong depression, almost entirely resting		3-Total anorexia		3-Watery severe yellow diarrhea	
**Vitality Score—VS**					
Score	Heart rate (beats/min)	Respiration rate (breaths/min)	Muscle tone	Skin color on the snout	Standing on all four legs
0	Absent	Absent	Flaccid	Pale	Absent
1	<120 (Bradycardia)	>40	Poor	Cyanotic	Poor
2	121–160 (Normal)	<20	Good	Red to pink	Good
3	121–160 (Normal)	20–36	Very good	Pink	Very good
4	>161 (Tachycardia)	20–36	Very good	Pink	Very good

**Table 3 vetsci-10-00135-t003:** Average Daily Weight Gain (ADWG) body weight values are presented by a mean and standard error on each recording period.

Group	D1 to D7	D8 to D14	D15 to D25
Group A-Cont	0.17 (0.02) ^abc^	0.20 (0.02)	0.31 (0.02)
Group B-Dexa1	0.10 (0.06) ^a^	0.15 (0.15) ^b^	0.29 (0.01)
Group C-Dexa3	0.08 (0.03) ^b^	0.17 (0.08) ^c^	0.27 (0.03)
Group D-Dexa + VitE/S1	0.19 (0.08)	0.20 (0.05)	0.32 (0.07)
Group E-Dexa + VitE/S3	0.25 (0.07) ^c^	0.27 (0.09) ^bc^	0.26 (0.05)

^a, b, c,^ statistically significant difference in compared pairs of groups in each period/column.

**Table 4 vetsci-10-00135-t004:** Mean body weight values are presented by a mean and standard error on each recording day.

Group	D1	D7	D14	D25
Group A-Cont	0.93 (0.01)	2.08 ^ab^ (0.09)	3.52 ^ab^ (0,13)	6.96 ^ab^ (0.10)
Group B-Dexa1	0.93 (0.02)	1.75 ^cde^ (0.11)	3.31 ^cde^ (0.15)	6.47 ^cde^ (0.12)
Group C-Dexa3	0.89 (0.10)	1.48 ^acfg^ (0.09)	2.75 ^acfg^ (0.13)	5.69 ^acfg^ (0.10)
Group D-Dexa + VitE/S1	0.88 (0.01)	2.24 ^dfh^ (0.09)	3.63 ^dfh^ (0.13)	7.10 ^dfh^ (0.10)
Group E-Dexa + VitE/S3	0.876 (0.01)	2.65 ^begh^ (0.09)	4.47 ^begh^ (0.13)	7.41 ^begh^ (0.10)

^a, b, c, d, e, f, g, h^ statistically significant difference in compared pairs of groups in each day/column.

**Table 5 vetsci-10-00135-t005:** Mean ranks of vitality score (VS) and post-hoc analysis [*p* and Standard error (std er)].

VS	Group	N	Mean Rank	Group A	Group B	Group C	Group D	Group E
D_1_	A	20	70.30		*p* = 0.01 std er = 8.0	nsd	**nsd**	nsd
	B	20	38.20	*p* = 0.01 std er = 8.0		nsd	nsd	nsd
	C	20	52.90	nsd	nsd		nsd	nsd
	D	20	52.90	nsd	nsd	nsd		nsd
	E	20	38.20	*p* = 0.01 std er = 8.0	nsd	nsd	nsd	
	Total	100		nsd	nsd	nsd	nsd	nsd
D_2_	A	20	63.20		*p* = 0.00 std er = 7.2	*p* = 0.015 std er = 7.2	nsd	nsd
	B	20	22.08	*p* = 0.00 std er = 7.2		nsd	*p* = 0.00 std er = 7.2	*p* = 0.00 std er = 7.2
	C	20	40.23	*p* = 0.015 std er = 7.2	nsd		*p* = 0.013 std er = 7.2	*p* = 0.013 std er = 7.2
	D	20	63.50	nsd	*p* = 0.00 std er = 7.2	*p* = 0.013 std er = 7.2		nsd
	E	20	63.50	nsd	*p* = 0.00 std er = 7.2	*p* = 0.013 std er = 7.2	nsd	
	Total	100		nsd			nsd	nsd
D_3_	A	20	58.78		*p* = 0.03 std er = 7.5	*p* = 0.05 std er = 7.5	nsd	nsd
	B	20	31.78	*p* = 0.03 std er = 7.5		nsd	*p* = 0.03 std er = 7.5	*p* = 0.00 std er = 7.5
	C	20	32.55	*p* = 0.05 std er = 7.5	nsd		*p* = 0.004 std er = 7.5	*p* = 0.00 std er = 7.5
	D	20	70.40	nsd	*p* = 0.03 std er = 7.5	*p* = 0.004 std er = 7.5		nsd
	E	20	59.00	nsd	*p* = 0.00 std er = 7.5	*p* = 0.0 std er = 7.5	nsd	
	Total	100		nsd	nsd	nsd	nsd	nsd
D_4_	A	20	47.30		nsd	nsd	nsd	nsd
	B	20	36.90	nsd		nsd	*p* = 0.006 std er = 6.95	*p* = 0.006 std er = 6.95
	C	20	47.00	nsd	nsd		nsd	nsd
	D	20	60.65	nsd	*p* = 0.006 std er = 6.95	nsd		nsd
	E	20	60.65	nsd	*p* = 0.006 std er = 6.95	nsd	nsd	
	Total	100		nsd	nsd	nsd	nsd	nsd
D_14_	A	20	48.88		nsd	nsd	nsd	nsd
	B	20	35.63	nsd		nsd	*p* = 0.00 std er = 6.13	*p* = 0.00 std er = 6.13
	C	20	46.50	nsd	nsd		nsd	nsd
	D	20	60.75	nsd	*p* = 0.00 std er = 6.13	nsd		nsd
	E	20	60.75	nsd	*p* = 0.000 std er = 6.13	nsd	nsd	
	Total	100				nsd	nsd	nsd

D_1_: Day 1-1st day of life, D_2_: Day 2-2nd day of life, D_3_: Day 3-3rd day of life, D_14_: Day 14-14th day of life. nsd (no significant difference). Group A-Cont (control group), Group B-Dexa1 (piglets treated with IM dexamethasone on D_1_), Group C-Dexa3 (piglets treated with IM dexamethasone on, D_1_, D_2_, and D_3_), Group D-Dexa + VitE/S1 (piglets treated with IM dexamethasone and vitamin E/Se on D_1_), Group E-Dexa + VitE/S3 (piglets treated with IM dexamethasone and vitamin E/Se on D_1_, D_2_, and D_3_).

## Data Availability

Not applicable.

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
