# Peer review of "Effects of Injectable Administration of Dexamethasone Alone or in Combination with Vitamin E/Se in Newborn Low Birth Weight Piglets"

_vetsci, 2023, doi:10.3390/vetsci10020135_

Round 1

Reviewer 1 Report

Review - Effects of injectable administration dexamethasone alone or in combination vitamin E/Se in newborn low birthweight piglets.

vetsci-2184603

General comment

Interesting study on potential interventions for postnatal low birth weight piglets to increase their vitality and survival. The design would have been complete if a treatment with vit E alone was included. Now it cannot be separated from the effect of dexa? The manuscript is written clearly in general, however, the English needs improving and the Results section needs to be improved in terms of writing style.

Simple summary

L15-19 improve flow of sentences. For example L18: “Vitality scores were lower in piglets that were administered Dexa for 1 or 3 days (Group B – Dexa1 and group C – Dexa3), respectively.”

Abstract

L28 .. alone or in combination with vitamin E/Se…

L31-31 Body weight was recorded for all piglets on days …., and vitality score was recorded on days …

L32-33 What are health and growth data more than vitality score and BW?

L37 administration of dexa and vitamin combined…..

Introduction

L43-44 Most of the neonatal mortality occurs in the first days …

L55 do you mean ….with reduced vitality..?

L62-68 In my view viral infections are not really in scope here. Please limit introduction to discussion of risks associated with LBW. It is sufficient to mention PPRS in the M&M

M&M

L121 In total, …

L122-125 Not clear. Do you mean that “100 sows were selected each with at least 14 (liveborn) piglets” since 14 piglets were kept with the sows without having to cross foster?

L126 weaning age was 25 days

L129 was checked everyday

L130 The farrowing rooms were equipped with…

Results

This section is poorly written in general. Please rewrite.

Please say BW when you mean body weight, and not BW score. It is not a score.

Section 3.2 is poorly written. For example L 192-195, I suppose this should say: “ Throughout the trial, animals in group C had a lower BW compared to all other groups, with the mean difference being …, … ,…, and …, for A, B, D, and E, respectively.” Please also rephrase other sentences. By the way, on d1 there were no differences.

Table 3 has an unorthodox way of using superscripts to indicate significant differnces. Please use the conventional way with a, b, c, etc and use different superscripts when two means in a column are significantly different.

Figure 1 is a duplication of Table 3, one of them can be removed.

Table 4 is not clear. I would expect empty cells on the diagonal because you cannot compare treatments wit themesleves (A v A, B vs B, etc).

Discussion

L270 could improve their survival…

L279 in comparison with

L284 investigated the effects of injectable dexamethasone…

L329-331 LBW pigletsthat received only dexa, showed poorer growth compared to piglets that received an IM …

Author Response

General comment

Interesting study on potential interventions for postnatal low birth weight piglets to increase their vitality and survival. The design would have been complete if a treatment with vit E alone was included. Now it cannot be separated from the effect of dexa? The manuscript is written clearly in general, however, the English needs improving and the Results section needs to be improved in terms of writing style.

We greatly appreciate your review and suggestions to improve our MS final form, please find our response

Simple summary

L15-19 improve flow of sentences. For example, L18: “Vitality scores were lower in piglets that were administered Dexa for 1 or 3 days (Group B – Dexa1 and group C – Dexa3), respectively.”

AU: Thank you for your comments. Please check our response on the text L15-19.

Abstract

L28 .. alone or in combination with vitamin E/Se…

AU: Thank you for your comments. Please check our response on the text L27.

L31-31 Body weight was recorded for all piglets on days ., and vitality score was recorded on days …

AU: Thank you for your comments. Please check our response on the text L31-32.

L32-33 What are health and growth data more than vitality score and BW?

AU: Thank you for your comments. We removed the sentence.

L37 administration of dexa and vitamin combined…..

AU: Thank you for your comments. Please check our response on the text L37.

Introduction

L43-44 Most of the neonatal mortality occurs in the first days …

AU: Thank you for your comments. Please check our response on the text L43-44.

L55, do you mean …. with reduced vitality...?

AU: Thank you for your comments. We have corrected the sentence on the text L55.

L62-68 In my view viral infections are not really in scope here. Please limit introduction to discussion of risks associated with LBW. It is sufficient to mention PPRS in the M&M

M&M

AU: We appreciate your comments. We removed the paragraph. Please check our response on the text.

L121 In total, …

AU: Thank you for your comments. Please check our response on the text L127.

L122-125 Not clear. Do you mean that “100 sows were selected each with at least 14 (liveborn) piglets” since 14 piglets were kept with the sows without having to cross foster?

AU: Thank you for your comments. Please check our response on the text L129-137.

L126 weaning age was 25 days

AU: Please check our response on the text L136.

L129 was checked everyday

AU: Thank you for your comments. Please check our response on the text L141.

L130 The farrowing rooms were equipped with… 

AU: Please check our response on the text L142.

Results

This section is poorly written in general. Please rewrite.

Please say BW when you mean body weight, and not BW score. It is not a score.

AU: Thank you for your comments. Please check our response on the text L212.

Section 3.2 is poorly written. For example, L 192-195, I suppose this should say: “Throughout the trial, animals in group C had a lower BW compared to all other groups, with the mean difference being …, … …, and …, for A, B, D, and E, respectively.” Please also rephrase other sentences. By the way, on d1 there were no differences.

AU: Thank you for your comments. Please check our response on the text L211-223.

Table 3 has an unorthodox way of using superscripts to indicate significant differences. Please use the conventional way with a, b, c, etc. and use different superscripts when two means in a column are significantly different.

Figure 1 is a duplication of Table 3, one of them can be removed.

AU: Thank you for your comments. We corrected Table 3. We totally agree with your comments for Figure 1, but with graphical representations, the reader can more easily understand the differences/effects

Table 4 is not clear. I would expect empty cells on the diagonal because you cannot compare treatments wit themselves (A v A, B vs B, etc.).

AU: Thank you for your comments. We are terribly sorry for this inattention. Please check our response on the text.

Discussion

L270 could improve their survival…

AU: Thank you for your comments. Please check our response on the text L307.

L279 in comparison with

AU: Thank you for your comments. Please check our response on the text.

L284 investigated the effects of injectable dexamethasone…

AU: Thank you for your comments. Please check our response on the text.

L329-331 LBW piglets that received only dexa, showed poorer growth compared to piglets that received an IM …

AU: Thank you for your comments. Please check our response on the text L321-322.

Reviewer 2 Report

The introduction is logically structured, easy to follow, and with relevant references. Maybe the section L48-61 could be improved by addding new reference(s) to published paper on hypothermia (now stated without reference in L59). 

Material and Methods

Please specify, detail of the injection procedure (e.g. section 2.2), as not presented. What type of syringe, needle (G, Size)? Was it changed between piglets or reused?

Please specify, the handling of each piglets, how was I done (entering the pen, one litter at a time during weighing).

Please specify, how long did the procedure of handling/weighing take – i.e. piglet away from the sow.

Please specify time of day of first intervention. How was day of birth (D0) determined? Relying on farm data, then observed when? This may introduce considerable variation in the data (piglet age), in particularly of importance during the early period. Or did you actually observe (video) the birth of first or last live piglet, and then adjusted Day 1 accordingly (a more precise  but cumbersome procedure).

Table 1. Please explain the treatment group more here. For example that the control group is injected with saline. Other control groups could have been included, e.g. no injection (as IM injection in itself may induce pain = a stressor).

These information are needed to (1) replicate your study methods, and (2) interpret your findings.

L131 please specify the temperature range, as relevant.

An important issue (L133-L136): The litter effect. How did you design the study? Was I one piglet per 100 litter, or could several LBW piglets be within the same litter? And if so, did you control/randomize the distribution of groups into a litter with more than one piglet entering the study? Please explain. In the statistics the litter effects needs to be taken into account, as piglets within a litter not are independent observations (random effect may be used). But this depend on your detailed study design.

Besides the litter effect, how did you do to randomise your treatment groups over time and parity? I guess, that you did not have access to 100 LBW piglet at once. Time of year and temperature may have an influence, and should be reported as well.

Further, you must include the sex of the piglet as well. In my opinion, it would be natural to include sex, parity and litter size at the time point of piglet weight for example. Potential influecing factos. The control of these factors are not presented and thus unclear in the current version.

LItter size is an important variable for the weight gain.

I cannot find a list of mortality data. 5/20 in one group died (25%), was it zero for the rest? Still in the simple summary L22, you conclude on survival rate, with no data on this.

L137 Were the experimenters blinded for the group during the data collection?

Table 2. Clinical Score 0, I suggest to change from ‘No abnormalities’ to ‘Appear normal’ in the first two columns.

What is ‘Physiological faces’? Please explain

Also explain how you create one score out of this. One, or all signs present for a given score, or rather, median of separate scoring of ‘General behaviour’, ‘Suckling’, ‘GI signs’? Not clear.

The first score goes from ‘good’ to ‘worst’ which is different from the next (from ‘worst’ to ‘good). Could be made more logical.

A score 0 piglet is a dead piglet? (No heart rate and respiration), so these are excluded from the data or included in some way?

For mortality, Why was these animals excluded? Clearly I understand that you did not weigh the dead ones, however, exclusion from the rest of the data (days when they lived) appear biased.

In case you calculated for example Average Daily Weight gain (ADW) per piglet, then these piglets could be included. I think is incorrect to remove these from for example the VS score. Thus you only report on the survivors?

3.2. L184. BW is not a score? Did you control for litter effect. Here also if some piglets are in a litter with fewer piglets and an older sow (parity 2 vs 1), we would expect a higher growth rate. As no cross fostering were use, you did not have e.g. 14 piglets for each litter. This needs to be taken into account in the statistical analysis.

L192. Should it be ‘Additionally, animals…’ or is it extra animals.

Table 3 and the statistical analysis. I suggest to use average daily growth per piglet, reducing the information to three periods (D 1-7, D7-14, D14-25). Not excluding piglets from earlier interval if dying later. Noting the number of piglets per group. Including sex, parity, litter size, litter effect in your analysis, to learn whether the effect reported really has something to do with the different treatments.

Figure 1. Please add units on you y-axis (kg) and x-axis (days 1-25). It would be appropiate with variations around these mean values, e.g. confidence intervals.

Figure 2 legend. Please explain what is meant with this Score. O is normal, 1-3 is intensively ‘worse’?

 Statistical analysis

As mentioned before, unclear how the litter effect was taken hand of. Other relevant co-factors could be sex (male, female) of piglet (were some even castrated – not described in the text, so apparently not?), parity of the sow (1, 2), actual litter size.

Discussion

L262-263 I suggest to delete ‘due to higher production needs’ as speculative. 

L272 unclear whether you mean ‘impair growth’ or ‘lack of effect’. I would like to see ADW per piglet reported.

L261 specify 'no statistically significant effect' rather than no effect.

L304 Authors should be more cautious with this proposal here. We do have an unhandled control group to compare with, and what if we left these piglets alone (or did other things to help them: several research papers point at other management procedures). Also the randomisation of treatment must be clearly explain, for us to beleive in the treatment effects.

The same points cautions should be stated in L 333-335 and in L344.

Alternativel solution, use genotypes with fewer but heavier, more vital piglets born?

It is rightfully acknowledged by the authors that Vitamin E/Se should be further investigated - as for now counfounded with Dex.

Minor points

Title: Should it be ‘administration of..’ ?

Simple Summary L16, S3 is introduced without reader knowing the meaning of this (Vit E/Se treatement for 3 days). The reference to the other groups not introduced, so this summary could be improved. L22 Survival rate not studied or reported, please specify in accordance to findings.

Author Response

The introduction is logically structured, easy to follow, and with relevant references. Maybe the section L48-61 could be improved by adding new reference(s) to published paper on hypothermia (now stated without reference in L59). 

AU: Thank you for your comments. Please check our response on the text.

Material and Methods

Please specify, detail of the injection procedure (e.g., section 2.2), as not presented. What type of syringe, needle (G, Size)? Was it changed between piglets or reused?

Please specify, the handling of each piglet, how was I done (entering the pen, one litter at a time during weighing).

Please specify, how long did the procedure of handling/weighing take – i.e., piglet away from the sow.

Please specify time of day of first intervention. How was day of birth (D0) determined?

AU: Thank you for your comments. Please check our response on the text L115-174

Each pig was injected in the right neck muscle intramuscularly, using an automatic syringe with a 20-gauge x ½ inch needle.

On the days of handing administration of Dexa alone or in combination with Vit E/Se, LBW piglets were placed in a transport cart and administered their assigned treatments. Separation of the LBW piglets from their sow and littermates lasted ~2–3 min. LBW piglets of each group were weighed using a digital weighing machine during each time point of the trial period. Separation of the LBW piglets from their sow and littermates lasted ~5–10 min.

The day of the first intervention (D1) for LBW piglets was the day of the farrowing (estimating time 0-9 after the birth of their last littermate).

Relying on farm data, then observed when? This may introduce considerable variation in the data (piglet age), in particularly of importance during the early period. Or did you actually observe (video) the birth of first or last live piglet, and then adjusted Day 1 accordingly (a more precise but cumbersome procedure).

AU: Thank you for your comments. Please check our response on the text L127-130 / L168-174.

In total, one hundred (100) newborn LBW piglets were selected from 100 sows (50 sows with parity 1 and 50 sows with parity 2) during the first 0-9 hours after the birth of the last piglet in each farrowing. The day of the first intervention (D1) for LBW piglets was the day of the farrowing (estimating time 0-9 after the birth of their last littermate).

Table 1. Please explain the treatment group more here. For example, that the control group is injected with saline. Other control groups could have been included, e.g., no injection (as IM injection in itself may induce pain = a stressor). This information is needed to (1) replicate your study methods, and (2) interpret your findings.

AU: Thank you for your comments. Please check our response on the text L151-156.

L131 please specify the temperature range, as relevant.

AU: Thank you for your comments. Please check our response on the text.

An important issue (L133-L136): The litter effect. How did you design the study? Was I one piglet per 100 litter, or could several LBW piglets be within the same litter? And if so, did you control/randomize the distribution of groups into a litter with more than one piglet entering the study? Please explain. In the statistics the litter effect needs to be taken into account, as piglets within a litter not are independent observations (random effect may be used). But this depends on your detailed study design.

AU: Thank you for your comments. Please check our response on the text L127-134.

In total, one hundred (100) newborn LBW piglets were selected from 100 sows (50 sows with parity 1 and 50 sows with parity 2) during the first 0-9 hours after the birth of the last piglet in each farrowing. Each selected sow had at least 14 (liveborn) piglets based on the farm records. In each selected sow total of 14 piglets were kept with the sow, while no cross-fostering was applied. LBW piglets were selected (1 piglet per litter) from newborn piglets on the first day of life based on their BW of less than 1 kg (range 0.6-1.0 kg).

Besides the litter effect, how did you do to randomize your treatment groups over time and parity? I guess, that you did not have access to 100 LBW piglet at once. Time of year and temperature may have an influence, and should be reported as well.

Further, you must include the sex of the piglet as well. In my opinion, it would be natural to include sex, parity and litter size at the time point of piglet weight for example. Potential influencing factors. The controls of these factors are not presented and thus unclear in the current version.

AU: Thank you for your comments. Please check our response on the text L126-138.

The study was carried out on a farrow-to-finish commercial pig farm with a capacity of 400 sows’ production (commercial hybrids of Large White × Landrace), located in southern Greece from April to May 2021. The capacity of the unit allowed us to complete the study in a short period of time with approximately the same conditions throughout the study

In total, one hundred (100) newborn LBW piglets were selected from 100 sows (50 sows with parity 1 and 50 sows with parity 2) during the first 1-6 hours after the birth of the last piglet in each farrowing. Each selected sow had AT LEAST 14 (liveborn) piglets based on the farm records. Therefore, the litter effect had a negligible to zero effect on the variables. Randomization was applied by the sealed envelope system when a sow with at least 14 piglets was presented.

Litter size is an important variable for the weight gain.

AU: Thank you for your comment. In our study, we selected sows with parity of at least 14 piglets

I cannot find a list of mortality data. 5/20 in one group died (25%), was it zero for the rest? Still in the simple summary L22, you conclude on survival rate, with no data on this.

AU: Thank you for your comment. No other loss of piglets was recorded except from Group B. We rephrase line 22

L137 Were the experimenters blinded for the group during the data collection?

AU: Thank you for your comments. Please check our response on L164-167.

The data were recorded by the same person. The vitality score and clinical observations of each piglet were scored continuously by the observer for the first 15 min of every hour of data collection. The observer was blinded as to time point, litter, and piglet treatment.

Table 2. Clinical Score 0, I suggest to change from ‘No abnormalities’ to ‘Appear normal’ in the first two columns.

What is ‘Physiological faeces? Please explain

Also explain how you create one score out of this. One, or all signs present for a given score, or rather, median of separate scoring of ‘General behaviour’, ‘Suckling’, ‘GI signs? Not clear.

AU: You are totally right we tried to make it more clear in a revised table 2. One sign was a score and further analyzed each time

The first score goes from ‘good’ to ‘worst’ which is different from the next (from ‘worst’ to ‘good). Could be made more logical.

AU: You are totally right but this is the way similar records were obtained according to several other studies.

A score 0 piglet is a dead piglet? (No heart rate and respiration), so these are excluded from the data or included in some way?

AU: Thank you for your comments. The term “Physiological faeces was replaced by “appear normal” which indicates the lack of any indication of digestive dysfunction. We observed dead piglets (score 0) on day 7 and animals on day 14 of the trial. Τhe data of dead animals were used in the analysis until the day of their death Please also check our response on the Table 2.

For mortality, why was these animals excluded? Clearly, I understand that you did not weigh the dead ones, however, exclusion from the rest of the data (days when they lived) appear biased.

In case you calculated for example Average Daily Weight gain (ADW) per piglet, then these piglets could be included. I think is incorrect to remove these from for example the VS score. Thus, you only report on the survivors?

AU: Thank you for your comment. You are absolutely right. The data of dead piglets were included in the study. Τhe verb «excluded» was misspelled after a poor translation of the corresponding Greek terminology.

3.2. L184. BW is not a score? Did you control for litter effect. Here also if some piglets are in a litter with fewer piglets and an older sow (parity 2 vs 1), we would expect a higher growth rate. As no cross fostering were use, you did not have e.g. 14 piglets for each litter. This needs to be taken into account in the statistical analysis.

AU: Thank you for your comments. We revise our presentation of eligible criteria in the text

L192. Should it be ‘Additionally, animals…’ or is it extra animals.

AU: Thank you for your comments. Please check our response on the text L192.

Table 3 and the statistical analysis. I suggest to use average daily growth per piglet, reducing the information to three periods (D 1-7, D7-14, D14-25). Not excluding piglets from earlier interval if dying later. Noting the number of piglets per group. Including sex, parity, litter size, litter effect in your analysis, to learn whether the effect reported really has something to do with the different treatments.

AU: Thank you for your comments. Please check our response on the text. We use the average daily growth. According to implied randomization and eligible criteria the above factor had negligible effect.

Figure 1. Please add units on your y-axis (kg) and x-axis (days 1-25). It would be appropriate with variations around these mean values, e.g. confidence intervals.

Figure 2 legend. Please explain what is meant with this Score. O is normal, 1-3 is intensively ‘worse’?

AU: Thank you for your comments. We revised our text .

 Statistical analysis

As mentioned before, unclear how the litter effect was taken hand of. Other relevant co-factors could be sex (male, female) of piglet (were some even castrated – not described in the text, so apparently not?), parity of the sow (1, 2), actual litter size.

In total, one hundred (100) newborn LBW piglets were selected from 100 sows (50 sows with parity 1 and 50 sows with parity 2) during the first 1-6 hours after the birth of the last piglet in each farrowing. Each selected sow had AT LEAST 14 (liveborn) piglets based on the farm records. Therefore, the litter effect had a negligible to zero effect on the variables. Randomization was applied by the sealed envelope system when a sow with at least 14 piglets was presented. Please check our response on the text L126-138.

Discussion

L262-263 I suggest to delete ‘due to higher production needs’ as speculative. 

AU: Thank you for your comments. Please check our response on the L300-301.

L272 unclear whether you mean ‘impair growth’ or ‘lack of effect’. I would like to see ADW per piglet reported.

AU: Thank you for your comments. We revised our text L308-312.

L261 specify 'no statistically significant effect' rather than no effect.

AU: Thank you for your comments. Please check our response on the text L318.

L304 Authors should be more cautious with this proposal here. We do have an unhandled control group to compare with, and what if we left these piglets alone (or did other things to help them: several research papers point at other management procedures). Also, the randomization of treatment must be clearly explained, for us to believe in the treatment effects.

AU: Thank you for your comments. No other intervention was applied in all groups of animals.

According randomization, the sealed envelope system was applied when a sow with a litter of 14 piglets was presented

The same points cautions should be stated in L 333-335 and in L344.

Alternatively, solution, use genotypes with fewer but heavier, more vital piglets born?

It is rightfully acknowledged by the authors that Vitamin E/Se should be further investigated - as for now confounded with Dex.

AU: Thank you for your comments. We made all appropriate changes on the text L366-371.

Minor points

Title: Should it be ‘administration of..’ ?

AU: Thank you for your comments. We corrected the title.

Simple Summary L16, S3 is introduced without reader knowing the meaning of this (Vit E/Se treatement for 3 days). The reference to the other groups not introduced, so this summary could be improved. L22 Survival rate not studied or reported, please specify in accordance to findings.

 AU: Thank you for your comments. We made all appropriate changes on the text L13-L22

Round 2

Reviewer 2 Report

I  have few further comments. Improvement suggested (figur 1) could inlude variation (e.g. 95% CI), and be made more readable than it currently is. However, the answers on methods from the authors were helpful.